# Postural Balance Effects Associated with 400, 4000 or 10,000 IU Vitamin D_3_ Daily for Three Years: A Secondary Analysis of a Randomized Clinical Trial

**DOI:** 10.3390/nu12020527

**Published:** 2020-02-19

**Authors:** Lauren A. Burt, Leigh Gabel, Emma O. Billington, David A. Hanley, Steven K. Boyd

**Affiliations:** McCaig Institute for Bone and Joint Health, Cumming School of Medicine, University of Calgary, 3280 Hospital Drive NW, Calgary, AB T2N 4Z6, Canada; lburt@ucalgary.ca (L.A.B.); leigh.gabel@ucalgary.ca (L.G.); emma.billington@ahs.ca (E.O.B.); dahanley@ucalgary.ca (D.A.H.)

**Keywords:** postural sway, sway index, clinical test of sensory interaction and balance, anterior-posterior, medio-lateral, aging, supplements

## Abstract

Vitamin D supplementation is proposed as a fall prevention strategy, as it may improve neuromuscular function. We examined whether three years of vitamin D supplementation (400, 4000 or 10,000 IU daily) affects postural sway in older adults. Three hundred and seventy-three non-osteoporotic, vitamin D-sufficient, community-dwelling healthy adults, aged 55–70 years, were randomized to 400 (n = 124), 4000 (n = 125) or 10,000 (n = 124) IU daily vitamin D_3_ for three years. Sway index was assessed at baseline, 12-, 24- and 36-months using the Biosway machine. We tested participants under four conditions: eyes open or eyes closed with firm (EOFI, ECFI) or foam (EOFO, ECFO) surfaces. Secondary assessments examined sway in the anterior-posterior (AP) and medio-lateral (ML) directions. Linear mixed effects models compared sway between supplementation groups across time. Postural sway under EOFO and ECFO conditions significantly improved in all supplementation groups over time. Postural sway did not differ between supplementation groups at any time under any testing conditions in normal, AP or ML directions (*p* > 0.05 for all). Our findings suggest that high dose (4000 or 10,000 IU) vitamin D supplementation neither benefit nor impair balance compared with 400 IU daily in non-osteoporotic, vitamin D-sufficient, healthy older adults.

## 1. Introduction

Muscle mass and bone density decline with age, increasing the risk for falls and fractures. Falls are a major health burden in the aging population, as they can result in fractures, restricted mobility, emergency room visits, admission to nursing homes and mortality. Thus, preventing falls is an important public health priority. Fall prevention programs that focus on improving balance, gait and muscular strength reduce the risk of falls by up to 27% [1]. Similar reductions in falls have been observed in individuals 65 years or older supplemented with ≥700 IU daily vitamin D [2].

Older adults supplemented with vitamin D demonstrate reduced risk of falls compared with non-vitamin D supplemented controls [2,3,4,5], though not all studies found a beneficial effect of supplementation on postural stability [6,7]. Muscle strength [5]; muscle function, including timed up-and-go [5,8] and chair stands [9]; balance [3,4,5,8]; reaction time [4] and gait [8] improved following vitamin D supplementation. However, positive studies often targeted vitamin D-deficient individuals [8] with a history of falls [3,4,9] and examined doses <4000 IU daily for <three years. Conversely, randomized clinical vitamin D supplementation trials demonstrated no effect of 800 IU supplementation in older women with a history of falls [9] and a detrimental effect of 2800 IU supplementation in older vitamin D-insufficient women on muscle function and postural stability [6,7]. The effects of long-term daily high-dose vitamin D supplementation on muscle function in healthy older adults is unclear.

In this study, we investigated the effects of three-years of vitamin D supplementation (400, 4000 and 10,000 IU daily) on postural balance (sway index) in community-dwelling older adults. Balance and measures of physical function were predefined secondary outcomes of our trial [10]. As previously reported [11], we did not find a relationship between high-dose vitamin D supplementation and changes in physical function (timed-up-and-go and grip strength). We hypothesized a dose-response relationship between vitamin D supplementation and sway index, with higher vitamin D dosages expected to improve postural sway.

## 2. Materials and Methods

### 2.1. Study Design

Our three-year randomized, double-blind clinical trial was designed to investigate the effects of daily vitamin D supplementation on bone density and strength [10] (Clinical Trial Registration NCT01900860). Secondary outcome measures included bone microarchitecture, balance, physical function and quality of life. We recruited participants from the general population by means of letter, posters and public media. A total of 542 individuals were screened; 373 met inclusion criteria, of which, the first 62 participants were enrolled in a pilot cohort (because of technical problems, the first 62 subjects did not receive baseline bone measurements on our second-generation HR-pQCT scanner but agreed to stay in the study, being eligible for all of the secondary outcome measurements) and 311 were enrolled in the main cohort [11]. The present study is an evaluation of balance outcomes. As the pilot cohort completed all balance assessments, they were included in this secondary analysis. We included men and women between the ages of 55–70 years, with women at least five-years post-menopause, residing near Calgary, Canada. We screened men and women for lumbar spine and hip bone mineral density and included participants if their T-score was above the threshold for osteoporosis (greater than −2.5). We did not exclude participants with chronic illness if the condition was stable and managed by a physician. We excluded participants if (1) their screening serum 25(OH)D was <30 nmol/L or >125 nmol/L; (2) serum calcium was >2.55 mmol/L or <2.10 mmol/L; 3) they consumed vitamin D supplements > 2000 IU/day for the past 6 months; (4) they were taking bone-active medication within the last two years; (5) they were diagnosed with disorders known to affect vitamin D metabolism, such as sarcoidosis, renal failure, malabsorption disorders or kidney stones, within the past year or (6) they regularly used tanning salons. The trial was approved by the Conjoint Health Research Ethics Board at the University of Calgary and Health Canada. Each participant provided written informed consent before randomization.

### 2.2. Randomization and Intervention

Participants were randomized in a 1:1:1 ratio to receive either 400, 4000 or 10,000 IU vitamin D_3_ cholecalciferol, taken orally once per day. A statistician unrelated to the trial generated a randomization table, which was uploaded into the study database by the database developers. To ensure the allocation of participants into study arms was blinded to all participants and study staff, the randomization table was only visible to the database developers. Health Canada (and the Institute of Medicine) recommend a daily vitamin D intake from all sources, for men and women aged 50–70 years, of 600 IU [12]. In this study, the lowest dose of 400 IU daily was chosen with the assumption that participants would receive at least 200 IU/day from diet.

The intervention ran for three years with the daily vitamin D_3_ supplementation taken orally in the form of liquid drops: five drops/day, dispensed in bottles to provide either three or six-month supplies at a time, given to participants at their scheduled laboratory and clinical assessment visits. Participants were provided calendars used as daily diaries to record vitamin D intake. At all visits, empty bottles were collected and counted, and diaries were checked to estimate adherence. The three doses were prepared by Ddrops Company, Woodbridge, ON, Canada. The concentrations of vitamin D varied according to the bottle (80 IU/drop, 800 IU/drop or 2000 IU/drop). Irrespective of group, participants ingested five drops/day. Quality control for each batch preparation of the tested doses of vitamin D (raw material testing, identification, assay of the three doses, testing for presence of heavy metals and microbiology) was carried out by three independent laboratories: Chemi Pharmaceutical Inc., Mississauga, ON, Canada; SGS Canada Inc., Mississauga, ON, Canada and Nutrasource Diagnostics Inc., Guelph, ON, Canada. Drop consistency was tested as part of quality control procedures, with a resulting variability of <3%. Participants were permitted to take up to 200 IU/day of additional vitamin D (e.g., a multivitamin supplement). In addition to vitamin D_3_ supplementation, each participant’s dietary calcium intake was assessed by a food frequency questionnaire [13]. Participants not consuming the recommended daily allowance (1200 mg/day) received additional calcium supplement tablets (300 mg elemental calcium as citrate) as needed, up to a maximum of 600 mg supplemental calcium/day.

### 2.3. Sample Size

The primary aim of this trial was to investigate the effect of vitamin D supplementation on bone density and strength [10]; thus, the size of the trial was powered for total bone mineral density (Tt.BMD) at the tibia or radius measured using high-resolution peripheral quantitative computed tomography. The trial was designed to have 90% power to detect a change in Tt.BMD between groups. Allowing for 20% attrition rate and using an alpha level of 0.025, we determined that 100 participants per group would provide 90% power to detect a difference between groups.

### 2.4. Descriptive Variables

We assessed height (m) and weight (kg) at each visit using standard methods [10] and calculated body mass index (BMI; kg/m^2^). Serum 25(OH)D (nmol/L) was measured using a chemiluminescent immunoassay (CLIA) evaluated on a DiaSorin Liaison XL system (DiaSorin, Stillwater, MN, USA), and performance was monitored using DEQAS quality assurance samples. Intra-assay coefficient of variance (CV) was 0.1%–3.8%, the inter-assay CV 6.0%–9.8% and the limit of detection (LOD) was ≤ 4.0 ng/mL (10 nmol/L).

### 2.5. Balance

We assessed sway index at baseline, 12-, 24- and 36-months using the Biosway machine (950–460, Biodex, NY, USA) [14]. Participants were instructed to stand on the Biosway platform with shoes removed with ten degrees of outward toe rotation and head in a neutral position. Participants completed the balance assessment under four testing conditions: (1) eyes open with firm surface (EOFI), (2) eyes closed with firm surface (ECFI), (3) eyes open with foam surface (EOFO) and (4) eyes closed with foam surface (ECFO). The foam (compliant) surface consisted of a 7.5-cm-thick piece of foam the same shape as the platform. This testing protocol is known as the modified Clinical Test of Sensory Interaction and Balance (m-CTSIB), without the visual conflict [15]. All tests were performed three times for 30 s without practice trials. Raw data were extracted from the Biosway machine. Prior to analysis, the first ten seconds was removed from each trial to allow for initial balance adjustments. We used the average of the three trials for each participant under each testing condition. Primary outcomes include indexes of postural sway (SWAY_EOFI_, SWAY_ECFI_, SWAY_EOFO_ and SWAY_ECFO_). Secondary outcomes include movement in the anterior-posterior (AP) and medio-lateral (ML) direction for each of the primary outcome variables. Sway index is the standard deviation of the stability index (average position from the center of gravity), with a higher score indicating greater instability during the test [14]. Sway index was calculated with a python script following exportation of raw data from the Biosway [16]. Data was passed through a Butterworth filter (5 Hz) prior to analysis. Reliability (ICC) of the Biosway ranges from 0.74 to 0.86 [17].

### 2.6. Statistical Analysis

Participant characteristics were described using means and standard deviations. The influence of vitamin D supplementation on balance variables across time was evaluated using linear mixed effects models. To determine the best-fitting model for all balance variables, an empty means random intercept model was fit to determine the amount of variance in sway indices attributed to between- and within-person differences. Second, a fixed linear time random intercept model was fit, followed by a random linear time model (allowing each participant his or her own slope for the effect of time). Wald test *p*-values were used to determine significance of individual fixed effects and maximum likelihood log likelihood (−2*log likelihood (LL)) statistics to determine significance of random effects variances and covariances between nested models given the difference in model degrees of freedom. A fixed linear time with a random intercept model was used for all variables except for SWAY_ECFO_, SWAY_ML_EOFI_, SWAY_ML_ECFO_, SWAY_ML_ECFI_ and SWAY_AP_ECFO_, where we used a random linear time model. Models included vitamin D treatment group and time as fixed main effects and a treatment by time interaction.

Model adequacy was checked graphically using plots of transformed residuals and adjusted means and estimated treatment differences in sway indices at each time were calculated using the margins command in Stata. A Bonferroni correction was used to account for multiple comparisons. Accordingly, the level of statistical significance was set to *p* < 0.017 (*p* < 0.05 divided by three) for group differences. We performed all analyses in Stata, Version 15.1 (StataCorp, College Station, TX, USA).

## 3. Results

### 3.1. Descriptive Characteristics

Of the 373 participants who were randomized, 345 participants remained at study competition (92% retention; Appendix A). For balance assessments, three participants were unable to complete the ECFO condition on one (n = 2) or all four (n = 1) occasions, and two participants were unable to complete the EOFO condition on one occasion. A summary of participant characteristics at baseline is shown in Table 1.

### 3.2. Vitamin D Supplementation Adherence and Serum Levels

Adherence with supplementation was 99.6% for 400 IU, 99.7% for 4000 IU and 99.1% for 10,000 IU. Baseline mean (SD) 25(OH)D levels were 76 (21), 80 (20) and 78 (18) nmol/L in the 400, 4000 and 10,000 IU groups, respectively. After three-months of supplementation, 25(OH)D levels were 76 (17), 114 (22) and 187 (38) nmol/L in the 400, 4000 and 10,000 IU groups. After thirty-six months of supplementation, 25(OH)D levels were 76 (18), 130 (27) and 142 (40) nmol/L in the 400, 4000 and 10,000 IU groups. Serum 25(OH)D levels in the 400 IU group did not change throughout the trial.

### 3.3. Sway Index

We summarize sway indices at baseline and absolute change from baseline at 12-, 24- and 36-months in Table 2. Figure 1 provides absolute change from baseline for sway index by vitamin D supplementation group, while Table 3 provides adjusted mean absolute differences between supplementation groups at each time point using linear mixed effects modeling. SWAY_EOFO_ and SWAY_ECFO_ demonstrated a significant time effect, such that indices significantly improved (sway decreased) over trial duration in all supplementation groups (Table 2). A summary of absolute change and adjusted mean differences between supplementation groups for sway in the AP and ML direction is in Appendix A.

There were no significant differences in sway indices between supplementation groups at any time under any testing condition in the normal, AP or ML directions (*p* > 0.05 for all).

## 4. Discussion

We examined the effect of three years of vitamin D supplementation on postural sway and found that it improved in the compliant, foam surface conditions over the three-year trial; however, this improvement was independent of vitamin D supplementation dose. Improved balance over time was likely a learning effect with the compliant foam surface condition rather than a supplementation effect. Specifically, participants in all treatment groups improved their postural sway on the foam surface over the trial, including those in our 400 IU group, whose serum vitamin D did not change with time [11]. In contrast, postural sway with the easier, firm surface condition did not improve over time and did not differ between vitamin D treatment groups.

Our findings are consistent with recent studies that assessed the dose-response relationship between high (6500 IU) and low (800 IU) [18] or medium (2800 IU) and placebo daily vitamin D supplementation and balance [7]. Although using different methodology to evaluate balance (tandem test and stadiometer), vitamin D supplementation did not affect balance in postmenopausal women who were vitamin D-sufficient with a low risk of falls at baseline [18] or in women with vitamin D insufficiency and hyperparathyroidism [7].

We highlight that participants in our study were healthy, community-dwelling adults who were vitamin D-sufficient at baseline without a history of falls; these may be the main reasons our findings differ from some previous research that found improved balance with vitamin D supplementation. Prior research examined vitamin D insufficient older adults [3,4,5,8] with a history of falls [3,4] and noted improved postural sway with up to 20 months of supplementation [3,4,5,8]. The high dosages (4000 IU and 10,000 IU) and three-year trial duration are unique to our study, as previous studies supplemented with lower daily vitamin D (range 400–1000 IU daily) and up to 24-months [5,6,8,19]. Importantly, we did not observe a detrimental effect of high-dose supplementation on balance, as has previously been reported in single-bolus delivery [20].

To further examine the specifics of postural sway, we decomposed postural sway into sway in the medio-lateral (ML) and anterior-posterior (AP) direction. Sway in ML [21,22,23,24] and AP [25] directions have been associated with risks for falls; however, the effect of vitamin D supplements on balance in these orientations is uncertain. Nine months of 1000 IU daily vitamin D supplementation has been shown to improve sway in both the ML (−37%) and AP (−36%) directions in vitamin D-insufficient postmenopausal women with a history of falls [3]. However, no improvement in ML sway was reported by another study supplementing 8400 IU weekly vitamin D for 16 weeks to vitamin D-insufficient older adults [26]. In our cohort of older healthy adults, we did not observe a dose-response effect in either ML or AP sway indices following daily vitamin D supplementation. Thus, there appears to be no advantage of disaggregating sway into ML and AP directions from the combined index.

Systematic reviews and meta-analyses exploring vitamin D supplementation report conflicting conclusions regarding the role of supplements for fall prevention. For example, vitamin D combined with calcium supplementation reduced the risk of falls [27], while vitamin D with or without calcium had no effect on falls [28]. In addition, a U-shaped daily dose-response relationship was reported where falls were not reduced with low-dose vitamin D (400 and 800 IU daily), falls were significantly reduced with medium-dose vitamin D (1600 to 3200 IU daily) and falls increased with high-dose vitamin D (4000 and 4800 IU daily) supplementation [29]. Specifically, individuals taking high daily dose vitamin D supplements were 5.6 times more likely to fall than individuals taking medium-dose vitamin D supplements [29]. Similar detrimental findings were observed in a randomized controlled trial (RCT), with falls and fractures following high-dose annual vitamin D supplementation [20] and in another RCT where 2800 IU daily reduced strength and functional measures of mobility compared with placebo [7]. Our study covered a wide range of daily supplementation doses, from 400 IU to 10,000 IU, and although it was not detrimental to the measured balance outcomes, it is also evident that the use of high-dose vitamin D supplementation did not improve balance in this healthy adult population.

Our study has several strengths and limitations. To our knowledge, this study is the first to look at changes in postural sway including two high doses of vitamin D supplementation (4000 and 10,000 IU daily) for three years. Further, 90% of participants completed our study, and adherence was >99%. Participants were healthy individuals with adequate serum 25(OH)D levels at baseline, without a history of falls and good balance at baseline, as demonstrated by sway index values lower than those previously reported for healthy men and women 65 to 84 years [14]. The above factors may have blunted sway-related changes observed over study duration. An important limitation of our study is that postural sway was a secondary outcome of the trial; thus, we may be underpowered to detect dose-response relationships in falls and sway indices, particularly because there is significant intra-participant variability. Further, unbeknownst to the investigators, two lots of the vitamin D preparations administered to the 10,000 IU daily group between months 18 and 36 suffered varying degrees of degradation, as discussed in more detail elsewhere [11]. The estimated delivered dose to these participants ranged between 2000 to 10,000 IU daily. Nevertheless, it is important to note that despite this unfortunate problem, the 10,000 IU group maintained the highest serum 25(OH)D levels across the trial. Due to the high doses of vitamin D administered during our study, participant safety was paramount. The results from our analysis concluded that high-dose vitamin D supplements are generally safe for participants; however, there were more cases of hypercalciuria and transient mild hypercalcemia among individuals in the 10,000 IU group [30]. Finally, participants may have differed in their consumption of vitamin D-rich foods and exposure to sunshine over the course of the study.

## 5. Conclusions

We did not observe a dose-response effect between vitamin D supplementation and postural sway. Our findings suggest that high-dose (4000 or 10,000 IU daily) vitamin D supplementation neither benefits nor impairs postural balance compared with 400 IU daily. Current recommendations of 400 IU daily [31] appear adequate for maintaining balance in non-osteoporotic, vitamin D-sufficient healthy, older adults without a history of falls.

## Figures and Tables

**Figure 1 nutrients-12-00527-f001:**
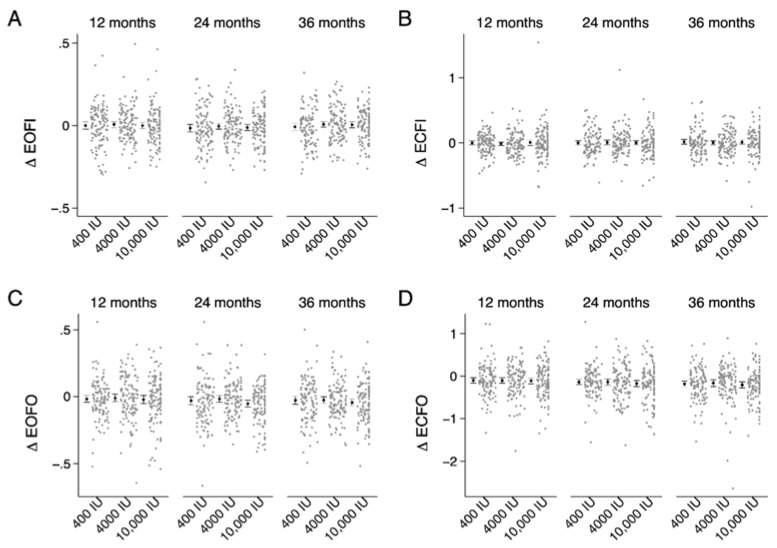
Absolute change in sway index from baseline by treatment group (400 IU, 4000 IU and 10,000 IU) across 36 months of vitamin D supplementation. Individual data points (grey) alongside group means and confidence intervals (black). (**A**) EOFI, eyes open firm surface; (**B**) ECFI, eyes closed firm surface; (**C**) EOFO, eyes open foam surface and (**D**) ECFO, eyes closed foam surface.

**Table 1 nutrients-12-00527-t001:** Participant characteristics at baseline.

Descriptive Variable	400 IU (*n* = 124)	4000 IU (*n* = 125)	10000 IU (*n* = 124)
Age (years)	62.0 (4.2)	62.7 (4.3)	62.0 (4.1)
Height (cm)	171.0 (9.0)	168.4 (9.2)	168.5 (9.5)
Weight (kg)	81.1 (15.1)	79.1 (15.7)	77.5 (15.0)
Body mass index (kg/m^2^)	27.7 (4.4)	27.8 (5.0)	27.2 (4.4)
Serum 25(OH)-vitamin D (nmol/L)	76 (21)	80 (20)	78 (18)
Total hip T-score	0.0 (1.1)	0.1 (1.2)	0.0 (1.1)
Falls ^a^ (%)	27 (21.8%)	22 (17.6%)	19 (15.3%)
Fracture since 50 years (%)	23 (18.5%)	16 (12.8%)	23 (18.5%)
History of cardiovascular condition (%)	24 (19.4%)	14 (11.2%)	16 (12.9%)
Type 2 diabetes (%)	3 (2.4%)	4 (3.2%)	5 (4.0%)
Rheumatoid arthritis (%)	2 (1.6%)	2 (1.6%)	1 (0.8%)
Asthma (%)	6 (4.8%)	10 (8.0%)	11 (8.9%)
Smoker (%)	3 (2.4%)	2 (1.6%)	5 (4.0%)

Data are mean (standard deviation) or n (%). ^a^ number of falls in the last 12 months.

**Table 2 nutrients-12-00527-t002:** Baseline and absolute mean change (SD) from baseline by treatment groups for sway index.

		SWAY_EOFI_	SWAY_ECFI_	SWAY_EOFO_	SWAY_ECFO_
Baseline	400 IU	0.4 (0.1)	0.7 (0.2)	0.7 (0.2)	2.2 (0.4)
	4000 IU	0.4 (0.1)	0.7 (0.2)	0.7 (0.2)	2.2 (0.5)
	10,000 IU	0.4 (0.1)	0.7 (0.2)	0.7 (0.2)	2.2 (0.5)
Δ 12-month	400 IU	0.00 (0.1)	0.00 (0.2)	−0.02 (0.1)	−0.10 (0.3) ^ad^
	4000 IU	0.01 (0.1)	−0.01 (0.2)	−0.01 (0.2)	−0.11 (0.4) ^ad^
	10,000 IU	0.00 (0.1)	0.01 (0.2)	−0.02 (0.2)	−0.11 (0.4) ^ad^
Δ 24-month	400 IU	−0.02 (0.1)	0.00 (0.2)	−0.03 (0.2) ^a^	−0.14 (0.3) ^a^
	4000 IU	0.00 (0.1)	0.01 (0.2)	−0.02 (0.1) ^a^	−0.14 (0.4) ^a^
	10,000 IU	−0.01 (0.1)	0.00 (0.2)	−0.05 (0.1) ^a^	−0.18 (0.4) ^a^
Δ 36-month	400 IU	−0.01 (0.1)	0.02 (0.2)	−0.03 (0.2) ^a^	−0.19 (0.4) ^ab^
	4000 IU	0.01 (0.1)	0.00 (0.2)	−0.03 (0.1) ^a^	−0.16 (0.4) ^ab^
	10,000 IU	0.00 (0.1)	0.01 (0.2)	−0.04 (0.1) ^a^	−0.21 (0.4) ^ab^

EOFI = eyes open firm surface, ECFI = eyes closed firm surface, = EOFO eyes open foam surface and ECFO = eyes closed foam surface. *p* < 0.05 (with Bonferroni adjustment) were significantly different from ^a^ baseline, ^b^ 12 months, ^c^ 24 months and ^d^ 36 months.

**Table 3 nutrients-12-00527-t003:** Adjusted mean absolute difference (95% confidence interval) in sway indices between treatment groups using linear mixed effects modeling.

		SWAY_EOFI_	SWAY_ECFI_	SWAY_EOFO_	SWAY_ECFO_
12-month difference	4000–400	0.004 (−0.03, 0.04)	0.005 (−0.06, 0.07)	0.015 (−0.03, 0.06)	0.063 (−0.08, 0.21)
	10,000–400	0.000 (−0.03, 0.03)	0.000 (−0.07, 0.07)	−0.008 (−0.06, 0.04)	0.008 (−0.14, 0.15)
	10,000–4000	−0.003 (−0.04, 0.03)	−0.005 (−0.07, 0.06)	−0.023 (−0.07, 0.03)	−0.055 (−0.20, 0.09)
24-month difference	4000–400	0.009 (−0.02, 0.04)	0.023 (−0.05, 0.09)	0.026 (−0.02, 0.08)	0.068 (−0.07, 0.21)
	10,000–400	0.005 (−0.03, 0.04)	−0.008 (−0.08, 0.06)	−0.022 (−0.07, 0.03)	−0.015 (−0.16, 0.13)
	10,000–4000	−0.004 (−0.04, 0.03)	−0.031 (−0.10, 0.04)	−0.048 (−0.10, 0.00)	−0.082 (−0.22, 0.06)
36-month difference	4000–400	0.012 (−0.02, 0.05)	0.005 (−0.06, 0.07)	0.013 (−0.04, 0.06)	0.089 (−0.05, 0.23)
	10,000–400	0.014 (−0.02, 0.05)	−0.011 (−0.08, 0.06)	−0.015 (−0.07, 0.04)	0.009 (−0.13, 0.15)
	10,000–4000	0.001 (−0.03, 0.03)	−0.015 (−0.08, 0.05)	−0.028 (−0.08, 0.02)	−0.080 (−0.22, 0.06)

EOFI = eyes open firm surface, ECFI = eyes closed firm surface, EOFO = eyes open foam surface and ECFO = eyes closed foam surface. No significant effect of group or group by time interaction.

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
