# Peer review of "Postural Balance Effects Associated with 400, 4000 or 10,000 IU Vitamin D3 Daily for Three Years: A Secondary Analysis of a Randomized Clinical Trial"

_nutrients, 2020, doi:10.3390/nu12020527_

Round 1
Reviewer 1 Report
It is a great opportunity to review the research article. This research article aimed to explore the effect of three years of vitamin D supplementation (400, 4000 or 10000IU daily) on postural sway among healthy community-dwelling older adults aged 55-70.
The advantages of the research article include excellent writing, sufficient background of relevant references, appropriate research design, clearly presented findings and the conclusions supported by the findings.
Although numerous advantages of the research article, there are some points need to be addressed clearly to merit the researchers and practitioners in the field.
As mentioned by authors, bone density, balance and muscle strength of lower limb were strongly associated the risk of falling among older adults. Please explain why the authors only select to measure the balance, not involved with muscle strength or bone density. Vitamin D can be produced sufficiently by human body by exposing to 10- or 15-minute sunshine daily. In addition, participant might consume vitamin D-rich food in daily diet. The confounding factors were not fully addressed in the research article. In the previous study, the dose of vitamin D supplementation report ranged from 400-8400 IU daily. Please provide the rationale of using 10000 IU for the third experimental group. There is an ethic issue need to be addressed. As mentioned by authors in the line 253-255 of the research article, 10000 IU daily group between months 18 and 36 suffered varying degrees of degradation. Is it appropriate to provide the high dose of 10000 IU daily to the participants? Moreover, did authors make what kind of effort to help solve the health problem of the suffering participants? Conflicts of Interest: Some authors received honoraria or research support from Amgen and the funding was provide by one foundation. Wold the sponsorship influence the independence of the study? Please address more to clarify.
Author Response
Author’s note: We thank the Editor and Reviewers for the opportunity to submit a revised version of our manuscript. We respond to each of the Reviewer’s comments below. We provide the page and line numbers of changes to the manuscript.
Response to Reviewer 1 Comments
Point 1: As mentioned by authors, bone density, balance and muscle strength of lower limb were strongly associated the risk of falling among older adults. Please explain why the authors only select to measure the balance, not involved with muscle strength or bone density.
Response 1: Thank you for this comment. The primary outcome of our three-year RCT was bone density and bone strength. We reported these results along with assessments of muscle function (grip strength and timed up and go) in our previous publication [Burt et al. (2019). Effect of high-dose vitamin D supplementation on volumetric bone density and bone strength: a randomized clinical trial. JAMA. 322(8):736-745]. There were no group differences in grip strength or timed up and go; however, the 10000 IU supplementation group had significantly lower volumetric bone density at the end of the trial.
Point 2: Vitamin D can be produced sufficiently by human body by exposing to 10- or 15-minute sunshine daily. In addition, participant might consume vitamin D-rich food in daily diet. The confounding factors were not fully addressed in the research article.
Response 2: Thank you for this comment. All participants were Vitamin D sufficient at baseline and there were no group differences in baseline Vitamin D status (Table 1). In Canada, many foods are fortified with Vitamin D and participants may produce sufficient vitamin D from sun exposure in summer. The randomization of our cohort at baseline ensured that sun exposure and vitamin D-rich food consumption was balanced between treatment groups. We added a sentence highlighting this as a limitation of the study on page 8, line 263.
“Finally, participants may have differed in consumption of vitamin D-rich food and exposure to sunshine over the course of the study.”
Point 3: In the previous study, the dose of vitamin D supplementation report ranged from 400-8400 IU daily. Please provide the rationale of using 10000 IU for the third experimental group. There is an ethic issue need to be addressed.
Response 3: When the study was designed, the Institute of Medicine had set the Tolerable Upper Intake Level (TUL) for Vitamin D at 4000 IU/day. However, there was persistent controversy as to whether the TUL (maximum dose deemed safe for the majority of the population could be raised to10,000 IU [Hathcock et al. (2007). Risk assessment for vitamin D. The American Journal of Clinical Nutrition, 85(1), 6–18 and Holick et al. (2011). Evaluation, treatment, and prevention of vitamin D deficiency: an Endocrine Society clinical practice guideline. The Journal of Clinical Endocrinology & Metabolism. 96(7),1911-1930].
The possibility that high vitamin D doses might have a beneficial effect on bone or other tissues had not been adequately explored, and at the time of initiation of the study, our research group had access to high resolution quantitative computed tomography, a technique that can assess subtle effects of vitamin D on bone microarchitecture (as we demonstrated in the JAMA paper mentioned above). Further, a study of high dose vitamin D in multiple sclerosis patients [Burton et al. (2010). A phase I/II dose-escalation trial of vitamin D3 and calcium in multiple sclerosis. Neurology, 74(23), 1852–1859] found no ill effects of a dose that averaged approximately 10,000 IU/day for 1 year.
Our study was approved by the Conjoint Health Research Ethics Board at the University of Calgary and received a letter of No Objection from Health Canada, as noted in our manuscript. Furthermore, the safety analysis of our study has recently been accepted for publication and is available online [Billington et al. (2019). Safety of high-dose vitamin D supplementation: secondary analysis of a randomized controlled trial. The Journal of Clinical Endocrinology & Metabolism. https://doi.org/10.1210/clinem/dgz212].
Point 4: As mentioned by authors in the line 253-255 of the research article, 10000 IU daily group between months 18 and 36 suffered varying degrees of degradation. Is it appropriate to provide the high dose of 10000 IU daily to the participants? Moreover, did authors make what kind of effort to help solve the health problem of the suffering participants?
Response 4: The result from our safety analysis concluded that high-dose vitamin D supplementation is generally safe for participants. While hypercalciuria and transient mild hypercalcemia occurred among individuals in all three groups, there were more cases in the 10,000 IU group. Cases of hypercalciuria and hypercalcemia resolved following a reduction in calcium intake.
The participants in our study were generally healthy or had stable health problems. We were not powered to assess changes in these general health problems over the study duration; however, these were reported in the above-mentioned safety publication. Specifically, we did not observe differences in risk of renal dysfunction, nephrolithiasis, falls or fractures between groups.
We have added a sentence and reference to our safety manuscript on page 8, line 260.
“Due to the high dose of vitamin D administered during our study, participant safety was paramount. The results from our analysis concluded that high-dose vitamin D supplementation is generally safe for participants; however, there were more cases of hypercalciuria and transient mild hypercalcemia among individuals in the 10,000 IU group.”
Point 5: Conflicts of Interest: Some authors received honoraria or research support from Amgen and the funding was provide by one foundation. Would the sponsorship influence the independence of the study? Please address more to clarify.
Response 5: The honoraria and/or research support from Amgen was for other research projects unrelated to this study and had no influence on the design, analysis or interpretation of data from this study. Funding for this study came from Pure North S’Energy Foundation and the funder had no role in study design, analysis or the interpretation of the data.
Reviewer 2 Report
This is clear and concise analysis of the effect of different vitamin D supplementation doses on balance improvement. The study is a very well performed and the study aim was clearly described.
Author Response
Response to Reviewer 2 Comments
The authors would like to thank the reviewer for their time and evaluation of our manuscript. There were no specific comments to address.
Reviewer 3 Report
This manuscript entitled "Postural balance effects associated with 400, 4000 or 10000 lU vitamin D3 daily for three years: a secondary analysis of a randomized clinical trial" describes a secondary analysis of the postural balance in a randomized clinical trial on 373 senior citizens that were daily treated with 400, 4000 or 10000 lU vitamin D3.
This pattern of supplementation with low, medium and high doses of vitamin D3 on an RCT results a novel approach to study the effect of vitamin D.
This paper is well-executed and clearly written.
Final standpoints are not mentioned (final 25(OH)D levels).
The RCT secondary analysis shows that the effect of vitamin D supplementation on postural sway has improved and that it is independent of vD doses.
This manuscript is worth to be opened to the vitamin D research community.
References should be checked carefully because the titles should be written as they appear in the article: for example:
7. “vitamin D3” should be written “vitamin D3” 7. The journal should be written without points 12. The journal should be written without points 15. Space is missing in “BalanceSuggestion”, it should be written “Balance Suggestion” 25. The journal should be written without points 26. “vitamin D(3)” should be written “vitamin D3” 27. The journal should be written without points 27. The journal should be written without points
Author Response
Author’s note: We thank the Editor and Reviewers for the opportunity to submit a revised version of our manuscript. We respond to each of the Reviewer’s comments below. We provide the page and line numbers of changes to the manuscript.
Response to Reviewer 3 Comments
Point 1: Final standpoints are not mentioned (final 25(OH)D levels).
Response 1: Thank you for this comment. We now report 25(OH)D levels on Page 5, line 169.
“After thirty-six months of supplementation, 25(OH)D levels were 76 (18), 130 (27), and 142 (40) nmol/L in the 400, 4000 and 10000 IU groups.”
Point 2: References should be checked carefully because the titles should be written as they appear in the article: for example:
- “vitamin D3” should be written “vitamin D3” 7. The journal should be written without points 12. The journal should be written without points 15. Space is missing in “BalanceSuggestion”, it should be written “Balance Suggestion” 25. The journal should be written without points 26. “vitamin D(3)” should be written “vitamin D3” 27. The journal should be written without points 27. The journal should be written without points
Response 2: Thank you for noting these errors. We have corrected all references.